# ADAPTIVITY AND MODULARITY FOR EFFICIENT GENERALIZATION OVER TASK COMPLEXITY

## ABSTRACT

Can transformers generalize efficiently on problems that require dealing with examples with different levels of difficulty? We introduce a new task tailored to assess generalization over different complexities and present results that indicate that standard transformers face challenges in solving these tasks. These tasks are variations of pointer value retrieval previously introduced by Zhang et al. (2021). We investigate how the use of a mechanism for adaptive and modular computation in transformers facilitates the learning of tasks that demand generalization over the number of sequential computation steps (i.e., the depth of the computation graph). Based on our observations, we propose a transformer-based architecture called Hyper-UT, which combines dynamic function generation from hyper networks with adaptive depth from Universal Transformers. This model demonstrates higher accuracy and a fairer allocation of computational resources when generalizing to higher numbers of computation steps. We conclude that mechanisms for adaptive depth and modularity complement each other in improving efficient generalization concerning example complexity. Additionally, to emphasize the broad applicability of our findings, we illustrate that in a standard image recognition task, Hyper-UT's performance matches that of a ViT model but with considerably reduced computational demands (achieving over 70% average savings by effectively using fewer layers).

## 1 INTRODUCTION

Tackling many real-world problems such as scientific research, math problem solving (Saxton et al., 2019; Wang & Lu, 2023), and parsing scenes (Kong & Fowlkes, 2018), requires reasoning over a chain of steps (Baldock et al., 2021; Agarwal et al., 2022) where the sequence of steps and the length of the chain are not known immediately. We conjecture that the basic ingredient to solve these learning problems in a generalizable and efficient manner is the capability to break up the learning problem into reusable components and compose them systematically (structured replacement of operations) and productively (constructing more complex operations by composing simpler ones) (Szabó, 2008).

An emergent technique to deal with this setting in Large Language Models (LLMs) is to use chain of thought (Wei et al., 2022a;b; Lee & Kim, 2023) where the chain of thought adjusts how much and what type of compute the model applies to solve a given example. *Can NNs learn a generalized chain of thought reasoning process without the need to tie it explicitly to their input/output?* Bubeck et al. (2023) argues that one of the limitations of LLMs such as GPT-4 is their inability to perform multi-step computations without an explicit scratch-pad (e.g., if not prompted to output the intermediate steps). For example, GPT-4 can correctly answer the question of counting prime numbers in a range only if it is asked to first list these numbers.

Mechanisms for having adaptive depth (Graves, 2016; Dehghani et al., 2019; Banino et al., 2021) and modularity/sparsity Perez et al. (2018); Ha et al. (2017); Pfeiffer et al. (2023) offer natural solutions to compose and execute computation graphs dynamically. Modularity helps with composing operations systematically and adaptivity enables constructing operations with varying complexities. Hence incorporating these mechanisms simultaneously into machine learning systems should increase their efficiency and performance in multi-step reasoning scenarios.

We design a synthetic task to probe the capability of models to generalize across complexity of examples in multi-step reasoning tasks. We aim to abstract away perceptual and other task-specific

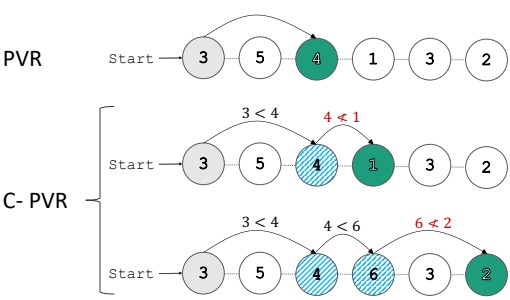

Figure 1: **Example of PVR and C-PVR Tasks.** In the regular PVR task, only one part of the input (in this case, the first element in the sequence) serves as the pointer. However, in the C-PVR, each element can act as a pointer. The task is to recursively retrieve the values until a condition is met. For this illustration, the condition is to continue as long as the sequence progresses forward. In this diagram, gray circles refer to initial steps, blue to intermediate steps, and green the final steps (the value of the green circle is returned).

factors that may otherwise lead to spurious correlations and other complexities not related to the central question of our work, and reduce task complexity to a number of sequential computation steps (referred to as hops) required to solve the task. This task, **conditional PVR** (C-PVR), is an extension of the pointer value retrieval task (PVR) (Zhang et al., 2021). We employ a symbolic sequential version of the task and introduce conditional multi-hop, meaning the number of steps required to solve an example is unpredictable until processed. In order to tackle this class of problems involving sequential reasoning steps, we extend the depth-adaptive framework of Universal Transformer (Dehghani et al., 2019) through the use of hyper-modules (Ha et al., 2017), resulting in what we call a **Hyper-UT** model. Hyper module is a hyper network (Ha et al., 2017) that composes a set of new weights via linear interpolation of weights from a bank of weight embeddings at each layer. Instead of sharing parameters across layers the hyper-network and the pool of weight embeddings are shared. This enables increasing the capacity of the model through increasing the number of modules while still benefiting from inductive biases of parameter sharing (Abnar et al., 2020; Tay et al., 2022), e.g., modular reuse.

Interestingly, we find that the inductive biases of adaptivity and modularity encoded in Hyper-UT are not only helpful in iterative reasoning tasks like C-PVR, but also in standard System-1 problems (Bengio et al., 2021) like Imagenet1K classification. In particular, we show that introducing the hyper-module component remedies the capacity problem of the Universal Transformer, and with this architecture, we can achieve the same level of performance on Imagenet1K benchmark while being more compute efficient.

As a summary, in this paper we make the following contributions:

- Introduce a new task to probe the capability of models to generalize across reasoning steps when the total number of steps is undetermined;
- We explore the interplay between adaptive depth mechanism and modularity and how they can synergize for efficient generalization in the context of example complexity;
- Examine the generality of the improvements in accuracy and efficiency of adaptive-depth, modular transformers on a standard image classification task;

## 2 MULTI-STEP REASONING

Multi-step reasoning is a process where a system breaks down the task into a sequence of steps and each step builds upon the previous ones. This type of problem solving is crucial in cases where there is ambiguity in the solution space, i.e., where the sequence of steps cannot be determined immediately. Scientific research is a prominent example of tasks that require such processes for forming a hypothesis and iteratively adjusting it based on the observations until a conclusion can be reached. A building block to enable a system to learn and execute this type of multi-step reasoning processes, is the capability to learn tasks that require iterations of dynamic length. We introduce a synthetic task, Conditional Pointer Value Retrieval (C-PVR) to probe this in neural networks (NNs).

## 2.1 Conditional Pointer Value Retrieval

We introduce C-PVR to investigate the capability and limits of NNs in performing tasks that require generalization over the number of sequential steps. The number of sequential steps can be viewed as a notion of complexity of examples. Generalization to length is a special case of this and it has been studied in both synthetic settings (Zhang et al., 2022; Abbe et al., 2023; Jelassi et al., 2023) and large scale setting (Anil et al., 2022). However, the number of sequential computation steps needed to solve the examples can be independent of their length (e.g., we can have shorter sequences that require more steps or longer sequences that require less). Additionally, we are interested in cases where there are no external or immediate clues for the model to know the sequence of steps or its length before solving the example.

Zhang et al. (2021) introduced PVR as a benchmark to study if transformers are capable of human-style reasoning. In PVR tasks, a specific portion of the input serves as a pointer, offering instructions that pertain to a particular input location. This location is then processed to generate the output. In a basic symbolic version of PVR tasks, inputs are sequences of numbers, where the first element in the sequence is the pointer to another element in the sequence, and the output value is computed based on the pointed element. In the simplest case, the output is the value where the pointer is pointing. In this task, the second step is conditioned on the output of the first step. A multi-hop version of this task can be constructed by allowing all the elements in the sequence to be interpreted as both pointers and values (directly or by applying some transformations) and specifying the number of hops at the example level or at the data-set level. We focus on the symbolic sequential version of this task and extend the notion of multi-hop PVR such that the number of steps is not given but depends on the input sequence. In simple terms, the task is defined as "Continue the retrieval steps recursively until a certain condition is met".

In designing the C-PVR task we follow two goals: (1) The minimum number of sequential steps should vary for different examples; and (2) The number of steps should not be identifiable before processing the sequence. The task comprises of the following components: (a) the input, which is a sequence of length $L$ of integers in the range of $(1, K)$, (b) the output which is an integer in the range of $(1, K)$, (c) the operation for getting the pointer from a given element, (d) the operation for getting the value from a given element, and (e) the halting condition.

In our experiments we set the halting condition to be get_pointer($a[i]$) > i (keep retrieving until we are not moving forward in the sequence). We compare two variants of our task: (1) **C-PVR (plain)** where both pointer and value functions are identity and $K = L$, and (2) **C-PVR (modulus)** where the value function is get_value($x$) = $x \% L$. Figure 1 shows examples of simple PVR and C-PVR (plain).

## 3 Adaptive and Modular Transformers

**Adaptive Compute** Given the aforementioned varying difficulty of examples many learning problems must tackle, a possible approach to address it is augmenting the model with an ability to dynamically allocate computational budget to different examples (or part of an example) accordingly. In this approach, a module, e.g., an LSTM in Graves (2016) or a transformer layer in Dehghani et al. (2019), is provided with an additional module (often a fully connected network) predicting a halting score that determines how many times this module will be repeated on the current input before a scoring threshold is met. To encourage the model to limit the amount of computation it will use, Graves (2016) proposes to not only truncate the maximum amount of updates, i.e., the number of repeated computations, but regularize the model using an additional Ponder Cost (for more details, see Graves, 2016).

**Modularity** Modularity is typically enforced in neural networks by breaking them into smaller sub-networks which can then be composed differently to solve different examples. Modular neural networks are interesting both for their potential generalization ability through a systematic composition of modules, their efficiency by reducing the cost of each compute step while allowing the total capacity of the model to grow, and sometimes their interpretability. There are different ways to make neural networks modular, e.g., mixture of expert transformer models (Fedus et al., 2022; Lepikhin et al., 2021), using hyper networks (Ha et al., 2017), modulating the activations (Perez et al., 2018), etc. We explore the following modularity techniques:

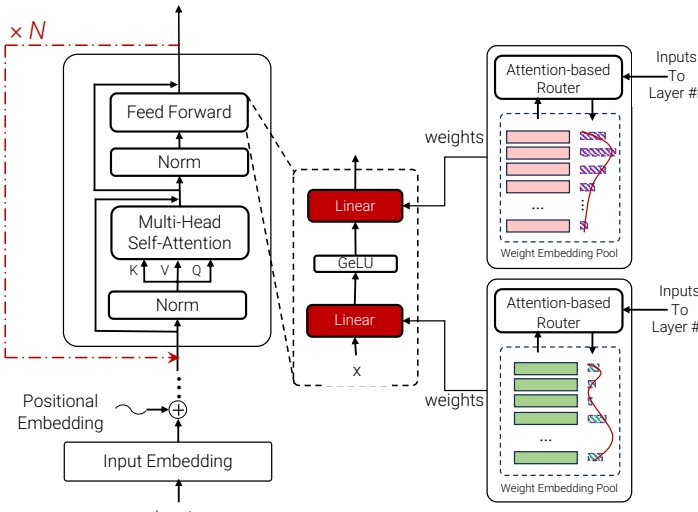

Figure 2: **Overview of the Hyper-UT Architecture.** In each Hyper-UT layer, the linear projection weights are dynamically generated by the "Module Selector", based on the input for each layer. Although the illustration primarily highlights the linear layers in the FeedForward block for simplicity, the same principle applies to the query, key, value, and output projections within the attention block.

- FiLM layers (Perez et al., 2018): The activations are modulated by shifting and scalar values that are predicted with an MLP block given varying input representations at each layer.
- Perceiver (Jaegle et al., 2021) style layers: At each step, there is an additional cross attention to a set of latent vectors where each vector can be interpreted as a module.
- Hyper-Module: The values of parameters of each layer are predicted by a hyper-module (that is shared across layers) given a representation of the input at each layer (Figure 2).

## 3.1 HYPER-UT

Universal transformers have been shown to generalize better than transformers in certain settings (Dehghani et al., 2019; Csordás et al., 2021); however, when applied to tasks which need higher capacity they often cannot match the performance of their non-adaptive counterparts (Xue et al., 2023). There are two main components for universal transformers: the adaptive depth module and the parameter sharing in depth. If the goal for applying adaptive depth is merely compute efficiency, one does not need to share parameters across depth and sacrifice capacity. But if the goal is to improve generalization, we want to jointly benefit from inductive biases of both adaptivity and parameter sharing (modular reuse). More importantly, if the goal is generalization, we expect models to be able to deal with examples of higher complexity at inference time.

The challenge of sharing parameters across the layers of the transformer model is that the only knob to increase the capacity of the model in terms of the number of parameters is to increase the width of the shared layer. This results in more flops per step or layer.

By incorporating the hyper-module into the UT architecture, we remedy this problem. While there is still an overhead for the modularization, we hypothesize that the capacity of the model can grow faster than the extra cost of compute per layer. Additionally, modularity can also be viewed as a source of inductive bias that can enhance both efficiency and generalization. Our experiments in Section 4.3 verify this to some extent.

In Hyper-UT, we replace the dense layers in the self-attention module and the MLP after the self-attention in the transformer block with hyper-modules (the hyper-modules for different components of the layers are not shared but they are shared across layers). At every layer, the hyper-module composes a set of new weights via linear interpolation of weights from a bank of weight embeddings, conditioned on some representation of the input at that layer. To avoid having to store weight embeddings of the size of the parameters, there is a projection layer that maps the module embedding to a vector which is the size of the weights needed to be predicted.

The Hyper-Module consists of the following components.

- Weight embedding pool: Each embedding in the weight embedding pool can be loosely interpreted as a module or expert. We split each embedding into key and value where the

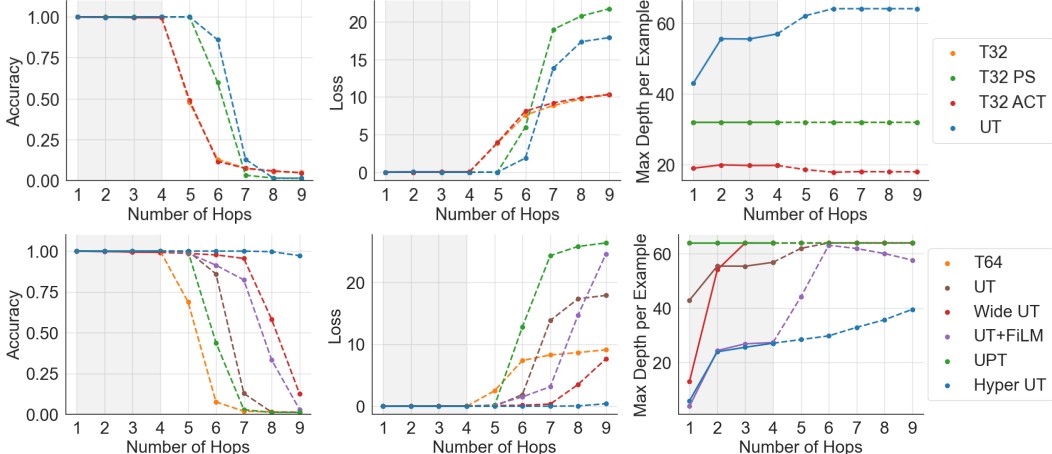

Figure 3: Performance of transformer variants on C-PVR (modulus) on test sets with different number of hops. The shaded area is for in-distribution and the un-shaded area is out-of-distribution. In the first row we examine the impact of adaptive depth and parameter sharing separately and when combined together. In the second row, we compare how different modularisation techniques improve the performance and efficiency of a model with adaptive depth. Note that in the first row, in the left most plot, T32 and T32 ACT overlap and in the right most plot, T32 and T32 PS overlap making T32 invisible in these two plots.

> key is used for module selection and the values are linearly interpolated based on a score generated by the router to compute the target module embedding.
> - Attention-based router: The attention-based router predicts the scores for each module to compute the embedding of the target module based on the attention between the representation of the input at the current layer and the keys in the weight embedding pool.
> - Weight generator: A linear layer that projects the weight embedding to the weights of the target module.

Since the weight generator needs to predict a vector with the size of the full matrix of the dense layers, it can become a bottleneck as we scale up the model size and increase the dimensions of these dense layers. To address this challenge, a potential solution is to predict factorized versions of the weight matrices. We defer the investigation of scalable implementations for the hyper-module to future research.

## 4 EXPERIMENTS

We present empirical results to show how incorporating adaptive compute and modular compute simultaneously yields better generalization and efficiency.

First, we show on the C-PVR task that generalizing to more complex examples (examples that require a higher number of sequential steps), is not trivial for standard transformers. Under a setting where the diversity of the complexity of examples seen during training is limited, we investigate the effects of adaptivity and modularity on the performance of transformers on this task.

Second, we investigate how pre-training on a language modelling task can help with learning and generalizing on the C-PVR task, and how augmenting the model with an explicit scratch-pad impacts the results.

Third, we look into a standard image classification task where models with adaptive depth do not match the performance of their non-adaptive counterparts despite being more efficient (Xue et al., 2023). In this setting, we show that introducing modularity, as we do in Hyper-UT remedies this problem, with Hyper-UT matching the performance of standard ViT (Dosovitskiy et al., 2021) with less compute in terms of the number of layers.

## 4.1 C-PVR: Efficient generalization over Example Complexity

In our experiments on the C-PVR task, we look into performance and efficiency of different adaptive and non-adaptive transformers with different modularization mechanisms. Our focus is on the generalization performance of these models with respect to the number of hops or the sequential computation steps needed to solve examples, as well as their efficiency in terms of correlation of the amount of compute and the complexity of examples.

We study the setting where the models are trained on examples of lower complexity (1-4) and evaluated on examples of higher complexity (5-9). Figure 3 shows the performance of the models on test sets with different numbers of hops in this setting. In this experiments, the sequence, $L$, is 100, and the values of elements are in the range of $[1, 1000]$. In the train set we have around $450k$ of each number of hops ($\sim 2M$ training examples) and the size of the test sets are $50K$.

We compare transformers with the following mechanisms:

- Adaptive Computation Time (ACT) (Graves, 2016) as the mechanism for having adaptive depth. We apply ACT per token.
- Different ways of parameter sharing and modularisation across layers with or without ACT:
  - No parameter sharing: The main problem with this approach is that if the model has adaptive depth and needs to be deeper during inference time, it needs to rely on parameters that are not well trained: T12 (Transformer with 12 layers), T32, T64, T32 ACT (Adaptive depth Transformer with max 32 layers)
  - Plain parameter sharing: All parameters are shared across all layers. We have one block that is called in a loop $N$ times, where $N$ is the number of layers: UT (Universal Transformer (Dehghani et al., 2019)), T32 PS (32 layer transformer with parameter sharing)
  - Parameter sharing with additional Film layers (UT+FiLM).
  - Sharing latents across layers of a perceiver style Transformer (UPT).
  - Hyper-Modules: Parameters of the dense layers are predicted by hyper-networks, which can be shared across layers: Hyper-UT

Details of the hyper-parameters of these models are presented in Appendix C.

**When do models break** Figure 3 shows how the accuracy of the models trained on examples with a smaller number of hops (1-4) drops as we increase the number of hops in the test set (from 1 to 9). We observe that the performance of the transformer models with no adaptive and modular compute drops more rapidly. Increasing the number of layers of a fixed-depth transformer model does not lead to any significant improvement in accuracy despite the increase in capacity (Figure 7 in Appendix A). On the contrary, parameter sharing in depth improves the generalization performance significantly, despite reducing the capacity in terms of number of parameters. In this particular setting adding an ACT mechanism without parameter sharing does not impact performance at all but it improves efficiency in terms of number of layers. UT, incorporating both parameter sharing and ACT at the same time, generalizes better than a standard transformer and a transformer with only parameter sharing. Adding modularity to UT in different forms, further improves both performance and efficiency. Among all, Hyper-UT achieves the best overall results. Moreover, increasing the width of the UT model, also leads to better generalization in terms of accuracy, verifying that a major problem for UT to generalize on this task is the capacity.

**Compute Efficiency and Fair Allocation of Compute** To investigate the compute efficiency of the different transformers we try on the C-PVR task, first, we look into the average depth of the models. Figure 3 shows how the average number of layers applied on examples varies for the test sets that require different numbers of hops. We expect a model to be efficient during inference not only if in general, it achieves better accuracy with less compute, but also if it can allocate compute to different examples not uniformly but based on their complexity.

For non-adaptive transformers average depth is always equal to the maximum number of layers. To make the comparison meaningful we train multiple instances of non-adaptive transformers with different numbers of layers (Figure 7) in Appendix A. We observe, in Figure 3, that UT+FiLM and Hyper-UT are more fair and efficient in terms of allocating compute compared to the UT model (HyperUT being the most efficient). By more fair in allocating compute, we mean the number of

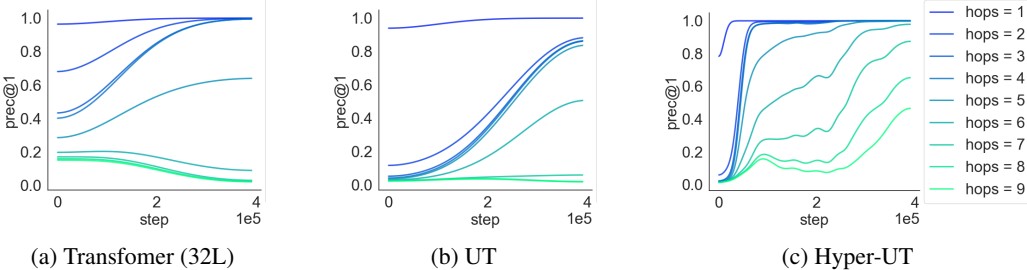

|  |  |  |
|---|---|---|
| (a) Transfomer (32L) | (b) UT | (c) Hyper-UT |

Figure 4: Implicit curriculum of transformers on C-PVR task. The y-axis is the accuracy and the x-axis is the training steps. This plot shows that transformers (a Hyper-UT instance) is learning examples in order of their complexity.

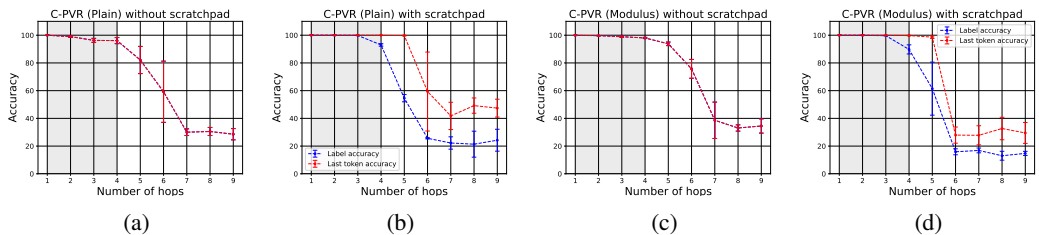

|  |  |  |  |
|---|---|---|---|
| (a) | (b) | (c) | (d) |

Figure 5: Performance of pre-trained T5 models on C-PVR (plain), (a) and (b) and C-PVR (modulus), (c) and (d). To generate each data point on the plots, we did three rounds of training. The reported values are the mean while the spread is just standard deviation of the sample mean.

effective sequential compute steps (e.g., number of layers), is correlated with the complexity of the examples. It is intriguing that complementing adaptive depth with modularity leads the adaptive models to use fewer number of layers (comparing UT with UT+FiLM and HyperUT).

**Implicit Curriculum** As depicted in Figure 4, the order of examples learned is correlated with their complexity (with respect to the number of hops). This means the notion of complexity we employ here is aligned with the family of solutions these models are learning.

## 4.2 PRE-TRAINED LANGUAGE MODELS

One natural question to ask is whether pre-trained language models are capable of generalizing to higher number of hops than those seen during training on the C-PVR (plain) and C-PVR (modulus) tasks using scratch-pad. To examine this question, we fine-tuned small pretrained T5 models on C-PVR (plain) and C-PVR (modulus) with or without scratch-pad. Note that scratch-pad could in principle allow models to allocate different amounts of compute to each input instance and potentially help them generalize to higher number of hops at inference time.

Figure 5 shows accuracy on C-PVR (plain)/C-PVR (modulus) tasks versus number of hops in two settings with and without scratch-pad. The T5 models are fine-tuned on an equal-sized mixture of examples with number of hops ranging from 1 to 4 (shown as the grey area in the plots). The models are tested on all number of hops from 1 through 9 (1 through 4 hops would be the in-domain test cases, while 5 through 9 hops would constitute out of distribution in our setting). The training set sizes are 100K/160K for C-PVR (plain)/C-PVR (modulus) respectively, while test size was fixed at 5K. All models are trained to convergence (C-PVR (plain)/C-PVR (modulus) for 9/15 epochs). The input size is fixed at $L = 30$. For C-PVR (modulus), the maximum of array elements value is 300. The problem is formulated as text-to-text. For runs with scratch-pad, the target has the general format of LABEL # SCRATCH-PAD, where scratch-pad is the string representing the array values at intermediate hops including the the label at the end. The character # is just a separator for convenience.

As we can see in Figure 5, fine-tuned T5 shows non-trivial generalization (e.g, well beyond chance-level) on number of hops higher than 4, which diminishes as the number of hops increases. Including a scratch-pad enhances the results if the accuracy is computed based on the last token of the scratch-

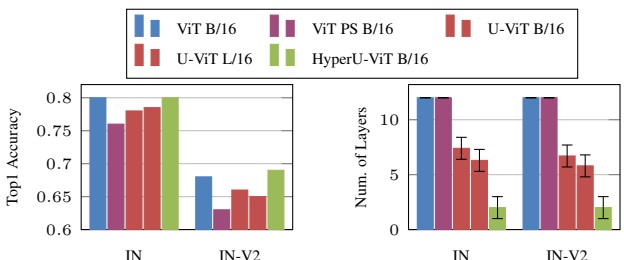

Average GFLOPs based on average number of layers on Imagenet1k validation set.

| Model | GFLOPs | Num. Layers |
|---|---|---|
| ViT B/16 | 17.76 | 12 |
| U-ViT B/16 | 10.51 | 7 |
| U-ViT L/16 | 15.89 | 6 |
| Hyper U-ViT B/16 | 3.45 | 2 |

Figure 6: Performance of vision transformer models trained on ImageNet1k on ImageNet v1 and ImageNet v2 validations sets. For details of hyper-parameters see Appendix D.

pad, but it hurts the performance if the accuracy is computed for the label that is followed by the scratch-pad. To get more insight into the type of errors made by the model with scratch-pad we look into the percentage of error where scratch pad and label are consistently wrong or are inconsistent. Figure-8 in Appendix B illustrates this. We observe that, for all number of hops across tasks, the dominate type of error is when model generates the exact scratch-pad (which contains the label at the end also) but fails to output the label itself (the first portion of the target string before #). As the number of test hops increases, the buckets where label is generated correctly but the scratch-pad is not an exact match to the ground truth emerges but remains small. Also, the bucket where neither label nor scratch-pad matches the ground truth becomes the second dominating type of error.

### 4.3 IMAGE CLASSIFICATION: CLOSING THE GAP BETWEEN U-ViT AND ViT

To show the generality of the benefits of combining adaptivity and modularity and the Hyper-UT architecture, we provide empirical results on a standard image classification task. In our experiments, U-ViT (a vision transformer model with parameter sharing and ACT mechanism) when trained on ImageNet1k, achieves a lower accuracy on this task compared to ViT (Xue et al., 2023). This is expected since U-ViT has a smaller number of parameters than ViT and hence smaller capacity. As mentioned earlier, Hyper-UT remedies the capacity problem of UT and we see in our experiments that Hyper-UT achieves the same level of performance as ViT and at the same time it converges to apply fewer layers on average. Interestingly, Hyper-UT converges to use fewer number of layers also compared to the U-ViT model. Additionally, we observe that increasing the capacity of a U-ViT model through some form of modularization (in this case hyper-module) has a more significant and robust effect compared to simply increasing the width of the model. Our findings are inline with the dissuasion in the concurrent work on sparse universal transformers (Tan et al., 2023).

Though the sheer number of layers is not an adequate measure of efficiency—given that the flops per layer vary across models—we note that, on average, Hyper-UT requires a lower total compute budget per example. This is despite the overhead of the hyper-module, which involves routing and weight prediction, because there is a substantial drop in the average number of layers used.

## 5 RELATED WORK

**Adaptive Compute** Adaptive compute has been a topic of interest in ML mainly because of its potential to lower inference costs (Laskaridis et al., 2021; Yang et al., 2020; Mehra et al., 2022; Hou et al., 2020). Other than its efficiency advantages, in some studies, it is shown that particular ways of incorporating adaptive compute introduce a form of recurrence in depth which can be a source of inductive bias that facilitates generalization in certain settings (Dehghani et al., 2019; Banino et al., 2021; Abnar et al., 2021; Csordás et al., 2021). RNNs, CNNs, and transformers are adaptive models that allocate compute to input based on the size of the input. While in principle all these models can deal with inputs of variable size their ability to generalize to inputs of varying shape is limited in their vanilla version (specific variants of RNNs, i.e., LSTMs are better at generalizing to unseen input sizes and in transformers using specific types of positional encodings could improve their ability to generalize to inputs of different size). However, the length/size of the input is not always a good proxy for its complexity. As a naive example, a given data point can be arbitrarily padded and resized but this does not increase its complexity. Graves (2016) introduced the concept of pondering and adaptive time compute and applied it to LSTMs. The main idea here is to allow the network to ponder as much as needed by having a module that predicts when to halt based on the current state, and

add a regularization term to the objective to minimize the number of steps. Later, Dehghani et al. (2019) applied the same ACT mechanism introduced by Graves (2016) to transformers, where you could make the halting decision per example or token. Banino et al. (2021) introduces a more stable mechanism for incorporating adaptivity by using a ponder loss function that is differentiable. In parallel, early-exit mechanisms (Kaya et al., 2019; Mehra et al., 2022), augment each layer with additional side branch classifiers such that if an example can already be classified in the lower layers it stops earlier. Alternatively, Xue et al. (2023) suggests Adatape that employs elastic input sequences to enable dynamic computation for example in transformers.

**Chain of Thought Reasoning** LLMs with a mechanism for chain-of-thought (Wei et al., 2022a; Nye et al., 2021; Wei et al., 2022b; Kojima et al., 2022) reasoning can potentially use the chain of thought as a strategy to control the amount of compute spent on each input example. It is shown that training language models with chain-of-thought increases their ability to generalize to longer sequences. There is however no study yet that investigates if merely allowing the model to adapt the amount of compute per example is the reason behind this success regardless of the content of the chain of thought, disentangled from other effects of prompt design (Brown et al., 2020).

**Modular and Sparse Compute** Modular compute, i.e., explicit or implicit sub-networks that are specialized and reused. Efficiency in terms of increasing the capacity of a model while not increasing the amount of compute per step and example is one of the main motivations for using modular NNs which is explored in a mixture of expert transformer models (Fedus et al., 2022; Lepikhin et al., 2021). Du et al. (2022) show that using a sparse mixture of expert transformers as the backbone for a language model, while increasing the number of parameters of the model, can lower the cost of inference while at the same time achieving better performance. Jaszczur et al. (2021) show that sparsity does not necessarily hurt the performance of the model if the total number of parameters is not increased. They show that sparse models achieve the same perplexity as the standard transformer with the same number of parameters. Additionally, modularity can potentially improve generalization by allowing the model to compose operations to deal with unseen data points (Goyal et al., 2021).

**Hyper-Networks** Ha et al. (2017) introduced hyper-networks, neural networks that predict the weights of the target model. They apply the idea on LSTMS, Hyper-LSTMS where every step, potentially a different set of weights can be applied on the next input token. Hyper-networks have been adopted to be used as modularisation techniques for efficient fine-tuning in transfer learning settings. It is also shown, theoretically, that hypernetworks can learn effective modular solutions (Galanti & Wolf, 2020). Mai et al. (2022) propose forming the token mixing MLP dynamically using hyper-networks in MLP-Mixers, achieving better performance than vanilla MLP-Mixers and better compute efficiency compared to transformers.

## 6 CONCLUSION

As a first step to assess and improve the capability of NNs for iterative problem solving, we propose a task to probe how well they can generalize on tasks that require dynamic number of steps per example. The task we propose, C-PVR, is an extension of the PVR task. In this task, we can split examples based on the (relative) number of sequential steps required to solve them. The task is designed such that the level of complexity or difficulty of the examples is not dependent on the length of the input. This has two advantages: (1) We can sidestep the challenges of length generalization related to input representation and positional embeddings. (2) This allows us to study the capability and efficiency of NNs to generalize over the complexity of examples without the side effects of the size (length) of the input on the computation budget of the models.

Our experiments provide evidence that in constrained settings, where the diversity of examples in the training set is limited, vanilla transformers struggle to generalize on the C-PVR task. We observe that pre-training transformers on language modelling improves their generalization in these settings. Additionally, mechanisms for adaptivity and modularity provide the models with inductive biases toward solutions that are more robust to the variations in depth of the computation graph needed to solve the examples. While adaptivity and modularity have been explored and employed separately in different settings to improve generalization and efficiency of NNs, here we show that they can have complementary roles and incorporating them into the model architecture at the same time can boost their effects. Moreover, we demonstrate that the advantage of integrating adaptivity and modularity extends beyond multi-step reasoning tasks, such as C-PVR. By simultaneously employing

these mechanisms, we can attain enhanced efficiency in handling a standard System-1 problem like ImageNet1k classification, all while maintaining accuracy.

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

## A    COMPARING TRANSFORMERS WITH DIFFERENT DEPTH

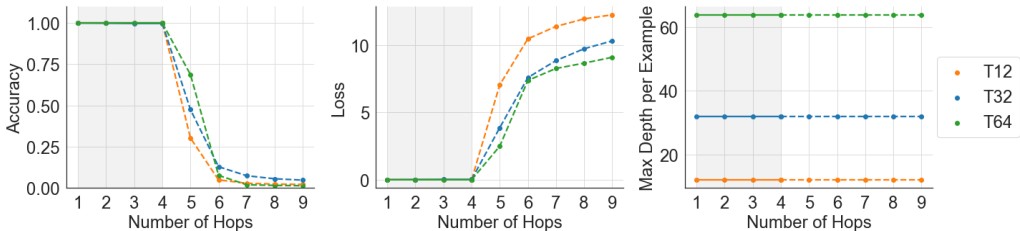

Figure 7: Performance of transformer variants on C-PVR (modulus) on test sets with different number of hops. We compare the performance of vanilla transformers with different number number of layers.

## B    ERROR ANALYSIS OF FINE-TUNED T5 MODELS WITH SCRATCH-PAD

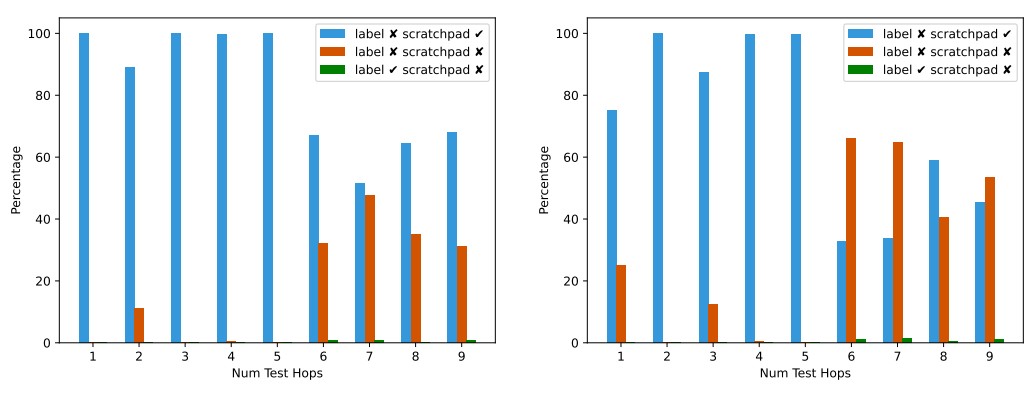

(a) Errors in C-PVR (plain) task with scratch-pad    (b) Errors in C-PVR (modulus) task with scratch-pad

Figure 8: Percentage of errors in scratch pad and label prediction for a T5 model fine-tuned on C-PVR (plain) and C-PVR (modulus). We observe that, for all number of hops across tasks, the dominate type of error is when model generates the exact scratch-pad (which contains the label at the end also) but fails to output the label itself (the first portion of the target string before #). As the number of test hops increases, the buckets where label is generated correctly but the scratch-pad is not an exact match to the ground truth emerges but remains small. Also The bucket where neither label nor scratch-pad matches the ground truth becomes the second dominating type of error.

## C    HYPER-PARAMETERS OF THE MODELS FOR THE EXPERIMENTS ON THE C-PVR TASK

All models are trained for 250 epochs with adam optimizer, a learning rate with cosine decay schedule and base learning rate of 0.001, and the weight decay of 0.0001. Details of hyper-parameter are listen in Table 1 and Table 2.

Table 1: Architectural details of transformer variants trained on C-PVR (modulus) from scratch.

| Model | Hidden Size | MLP dim | Num. Layers (max) | Num. Heads | Positional Embedding |
|-------|-------------|---------|-------------------|------------|----------------------|
| T12 | 384 | 1536 | 12 | 6 | Learned |
| T32 | 384 | 1536 | 32 | 6 | Learned |
| T64 | 384 | 1536 | 64 | 6 | Learned |
| UT | 384 | 1536 | 64 | 6 | Learned |
| UT Wide | 768 | 3072 | 64 | 12 | Learned |
| UT+FiLM | 384 | 1536 | 64 | 6 | Learned |
| UPT | 384 | 1536 | 64 | 6 | Learned |
| HyperUT | 384 | 1536 | 64 | 6 | Learned |

Table 2: Hyper parameters of adaptive transformer variants trained on C-PVR (modulus) from scratch.

| Model | Param. Sharing | Modular MLP | Modular Att-out | Modular Att-kqv | Router Temp | ACT Type | Num. Modules |
|-------|----------------|-------------|-----------------|-----------------|-------------|----------|--------------|
| UT | True | None | None | None | - | Token | - |
| UT Wide | True | None | None | None | - | Token | - |
| UT+FiLM | True | FiLM | FiLM | FiLM | - | Token | - |
| UPT | True | None | None | None | - | Token | 128 |
| HyperUT | True | Hyper Module | Hyper Module | Hyper Module | 1.0 | Token | $32 \times 128$ |

# D    HYPER-PARAMETERS OF THE MODELS FOR THE EXPERIMENTS ON IMAGENET1K CLASSIFICATION

All models are trained for 450 epochs with adam optimizer, a learning rate with cosine decay schedule and base learning rate of 0.001, and the weight decay of 0.0001. Details of hyper-parameter are listed in Table 3.

Table 3: Hyper parameters for experiments with ViT variants on the Imagenet1K classification task. The architectural hyper-parameters of the models (B/16) and (L/16) are the same as the ones reported in  (Dosovitskiy et al., 2021).

| Model | Param. Sharing | Modular MLP | Modular Att-out | Modular Att-kqv | Router Temp | ACT Type | Num. Modules |
|-------|----------------|-------------|-----------------|-----------------|-------------|----------|--------------|
| ViT B/16 | False | None | None | None | - | - | - |
| ViT PS B/16 | True | None | None | None | - | - | - |
| U-ViT B/16 | True | None | None | None | - | Token | - |
| U-ViT L/16 | True | None | None | None | - | Token | - |
| HyperUT B/16 | True | Hyper Module | None | Hyper Module | 1.0 | Token | $128 \times 256$ |

# E    ATTENTION BASED ROUTER

For the attention based router, we use a standard attention layer (multi-head cross attention) (Vaswani et al., 2017). In the experiments presented in this paper, we set the number of heads to be 1. In the attention based router, the query is the example embedding at the current layer, and the key/values are the embeddings in the module embedding pool.

# F    ADAPTIVE COMPUTATION TIME

We apply the ACT mechanism per token, and for the details of the method we refer to the original paper (Graves, 2016), and for how it is incorporated into the transformer architecture we refer to the universal transformer paper (Dehghani et al., 2019).

In this approach, there is a module (a MLP with a sigmoid activation), that given some representation of the input tokens at every layer predicts a halting score. Then at every layer, an accumulated halting score is calculated, and the model halts if this score is higher than $1 - \epsilon$.

$$h_t^n = \sigma(W_h s_t^n + b_h) \tag{1}$$

Then the objective of the model is augmented with the ACT loss, Equation 2, which is a penalty term that encourages the model to halt earlier, and an additional regularization term to make the halting score at the time of halting closer to one.

$$N(t) + (1 - \sum_{n=1}^{N(t)-1} h_t^n) \tag{2}$$

where $N(t)$ is the number of updates (steps):

$$N(t) = \min\{n' : \sum_{n=1}^{n'} h_t >= 1 - \epsilon\} \tag{3}$$

We could apply ACT per token or per example. The current experiments in the paper report the results for when ACT is applied per token. In our experiments the weight of the ACT loss is set to 0.1.

## G EXAMPLES OF THE C-PVR TASK

Here are a few exampels, of the C-PVR task we experiment with.

| Sequence | Label | Number of Hops |
|---|---|---|
| 65 29 36 16 23 69 99 75 2 72 61 12 13 74 10 54 | 65 | 1 |
| 94 40 22 79 56 50 2 68 81 88 77 43 6 81 20 75 | 81 | 2 |
| 39 70 96 62 23 42 41 21 45 81 54 44 43 66 44 79 | 43 | 4 |
| 20 90 20 24 63 62 19 27 5 63 28 31 90 54 29 59 | 29 | 6 |
| 38 31 12 33 76 39 40 48 56 42 73 80 49 26 20 44 | 44 | 5 |
| 49 87 35 7 36 64 74 3 3 79 76 59 57 27 77 25 | 49 | 1 |
| 21 72 25 67 24 2 80 20 63 53 87 22 66 24 32 74 | 20 | 3 |
| 19 25 20 25 80 36 24 65 26 20 62 27 100 98 28 35 | 20 | 5 |
| 3 25 7 72 65 75 9 8 12 72 86 8 31 46 50 61 | 8 | 5 |
| 82 87 26 18 60 99 92 52 86 64 34 96 45 13 20 88 | 88 | 5 |

Table 4: Examples from the instance of the C-PVR task used in our experiments (sequence length is 16).

Here we include the script for computing the label given the string for the the C-PVR task.

Listing 1: Retrieval function for C-PVR task

```python
def retrieval_rule(sequence, get_pointer_fn, get_value_fn, condition_fn):
    pointer = 0
    num_hops = 1
    next_pointer = get_pointer_fn(pointer, sequence)
    while condition_fn(pointer, next_pointer, len(sequence)):
        pointer = next_pointer
        next_pointer = get_pointer_fn(pointer, sequence)
        num_hops += 1

    return get_value_fn(pointer, sequence), num_hops
```

One can build different version of the C-PVR task by defining different `get_pointer_fn`, `get_value_fn` and `condition_fn`.

Listing 2: Auxiliary Functions for the instance of the C-PVR task we use in this paper

```python
def get_value(pointer, sequence):
    return sequence[pointer]

def get_pointer(pointer, sequence, sequence_length):
    return (sequence[pointer] - 1) % sequence_length)

def condition_fn(pointer, next_pointer):
    return pointer < next_pointer
```

