# OpenReview forum: "Adaptivity and Modularity for Efficient Generalization Over Task Complexity"
_ICLR.cc/2024/Conference — Submitted to ICLR 2024_

### Official Review · Reviewer_Qbwm · 2023-10-30

**Soundness:** 2 fair
**Presentation:** 2 fair
**Contribution:** 2 fair
**Rating:** 5
**Confidence:** 2

**Summary:**

This paper proposes a task to probe the capability of models to generalize across reasoning steps and the authors design a transformer-based architecture that combines dynamic function generation from hypernetworks with adaptive depth from universal transformers. Extensive experiments show the effectiveness of the proposed approach.

**Strengths:**

Exploring the general efficiency of transformers on problems that require dealing with examples with different levels of difficulty seems novel.

**Weaknesses:**

1. The task definition is unclear.
2. The proposed method seems a combination of transformers and hyper-modules (Ha et al., 2017).
3. In the experiment, the authors do not compare the state-of-art methods.

**Questions:**

see the weakness

---

> ### Author Response · Authors · 2023-11-20
>
> Thank you for reviewing our paper. Please have a look at our shared response. And here are our comments/questions about some of your remarks about the weaknesses of the paper.
>
> ### Task definition:
> We would appreciate if you could tell us which parts of the task definition you found unclear. We will do our best to clarify this as much as possible in the camera ready version (Please see appendix G in the updated version in updated version of the paper).
>
> ### Comparison with the state of the art:
> The focus of the paper has been to study the ability of transformers to generalize to example of higher complexity while allocating compute proportionally. Hence, our baseline is a vanilla transformer and we compare how augmenting the vanilla transformer with adaptive compute time and modularity enhances their generalization and efficiency. We think it makes sense to study transformers since they seem to be the main building block of current state-of-the-art models.  After showing that it is challenging for vanilla transformers to generalize in the specific setting of the C-PVR task, we show that adaptivity and modularity have complementary effects on improving the generalization and efficiency of transformer models.
> If you could clarify what are the other methods/models that you would like us to add to our comparisons, we will try to include them in the future versions of the paper.

---

### Official Review · Reviewer_D459 · 2023-10-31

**Soundness:** 4 excellent
**Presentation:** 2 fair
**Contribution:** 3 good
**Rating:** 5
**Confidence:** 3

**Summary:**

The paper proposes Hyper-UT which is a Transformer architecture that can perform adaptive computation and dynamically generate weights. The goal is to have Transformers that can generalize to higher number of computation steps and effectively allocate different amounts of computation depending on input complexity. The paper also introduces C-PVR task to measure the multi-step generalization of various Transformer models and found that existing Transformers perform worse than Hyper-UT. Hyper-UT also demonstrates strong accuracy and efficiency on ImageNet tasks, compared to ViT and U-ViT.

**Strengths:**

- Good analysis on the weaknesses of existing Transformers using the proposed multi-step reasoning task C-PVR.
- Intriguing finding that combining adaptive computation and modularity can have strong complementary effects.
- Experimental results are very promising, especially the HyperU-ViT results on ImageNet datasets. HyperU-ViT matches or outperforms traditional ViT using a significantly small fraction of FLOPS that ViT requires.

**Weaknesses:**

- The individual components like ACT and Hyper-Module are not new inventions. They have been introduced in existing papers. This paper merely uses the two within Transformer in a relatively simple manner.
- The details on how and where ACT is being used in Hyper-UT/HyperU-ViT is lacking. It is also similar for Hyper-Module. While there's diagram that shows the attention-based router, it is unclear how attention-based router is formulated (i.e., there's no equation).
- There's no comprehensive ablation study on the different components.

**Questions:**

Can this be applied to NLP Transformers (e.g., LLMs)? What's the reason of using ViT/image classification as the testbed?

---

> ### Author Response · Authors · 2023-11-20
>
> Thank you very much for the review and the interesting question. Please see our shared response. Also, here are the response to your question and clarification on your remark on the lack of details.
>
>
> `Lack of details for ACT and Hyper-Module in Hyper-UT architecture:`
>
> We will explain the ACT and the Hyper-Module and their corresponding hyper-parameters with more details in the appendix.
>
> **ACT mechanism:**
> We apply the ACT mechanism per token, and for the details of the method we refer to the original paper (Graves, 2016), and for how it is incorporated into the transformer architecture we refer to the universal transformer paper (Dehghani et al, 2019). Basically, there is a dense layer with sigmoid activation (ACT unit), that given the representation of each token at the current layer, predicts a score (the halting score). The model will halt at a given layer for a given token when the sum of all the halting scores for the token in the previous layer is bigger than $1 - \epsilon$. We could also do this per example, but the current experiments in the paper report the results for when ACT is applied per token.  Also, the objective of the model is augmented with the ACT loss, which is:
> <number of steps> + (1 - <final halting_score>).
>
>
> **Attention based router:**
> For the attention based router, we use a standard attention layer (multi-head cross attention, in our experiments with just one head). Where the query is the example embedding at the current layer, and the key/values are the embeddings in the module embedding pool.
>
> `Can this be applied to NLP Transformers (e.g., LLMs)? What's the reason of using ViT/image classification as the testbed?`
> Of course, we just wanted to show that the idea generally applies and can be useful even in system-1 like tasks where there is not much need to a lot of sequential processing. Hence we chose image classification. We agree it would be very interesting to see the results of applying these to LLMs. We might even see more benefits on language modeling since, it sounds like a much more complex task. While we have deferred this to future work, we would like to point out to a concurrent work that applies a similar idea to NLP tasks (Tan et al, 2023).
>
> [1] Shawn Tan, Yikang Shen, Zhenfang Chen, Aaron Courville, and Chuang Gan. Sparse universal transformer. arXiv preprint arXiv:2310.07096, 2023.

---

### Official Review · Reviewer_PAjo · 2023-11-06

**Soundness:** 3 good
**Presentation:** 3 good
**Contribution:** 2 fair
**Rating:** 5
**Confidence:** 2

**Summary:**

This paper proposes a new hypernetwork-structured parameterization for transformers, which focuses on adaptive and dynamic computation. In showcase its advantages, the paper considers a new task called conditional pointer value retrieval and demonstrates some improvement on this task and conventional ImageNet classification.

**Strengths:**

- The paper is well structured and clearly presented in general. It is easy to follow.

- The dynamic nature of the proposed Hyper-UT is interesting and makes intuitive sense, as for different task/input, different levels of computation complexity is required.

- The experimental results look good and Hyper-UT indeed shows some improvement.

**Weaknesses:**

- Hyper-UT is essentially an alternative parameterization of HyperNetworks, despite the dynmaic computation is placed in transformer blocks. I don't find it particularly different from HyperNetworks and its numerous variants, say [1,2,3,4,5].

- The motivation behind Hyper-UT is not clear to me. Although the design of HyperNetwork-like stucture makes the computation dynamics and may save the inference cost, I fail to understand the motivation of the design choices made in Hyper-UT. Why the proposed design will be better than many other HyperTransformers is not clear to me. It seems that it is simply yet another hypernetwork.

- The experiment on C-PVR is not particularly convincing, as the task is a synethic one rather than a real problem. Could the authors find some other tasks that are more representative and yet realistic?


[1] Revisiting Linear Decision Boundaries for Few-Shot Learning with Transformer Hypernetworks, openreview 2022

[2] On the Modularity of Hypernetworks, NeurIPS 2020

[3] HyperGrid Transformers: Towards A Single Model for Multiple Tasks, ICLR 2021

[4] Parameter-efficient Multi-task Fine-tuning for Transformers via Shared Hypernetworks, arXiv:2106.04489

[5] Hypermixer: An mlp-based low cost alternative to transformers, ACL 2023

**Questions:**

See the weakness section.

---

> ### Author Response · Authors · 2023-11-20
>
> Thank you for the feedback. We have responded to your comments about novelty and ablation experiments in the shared response.  Here are our comments on some of the remaining points:
>
> `“Hyper-UT is essentially an alternative parameterization of HyperNetworks...”`
> Please see the shared response, the section on the concerns over novelty. Also, thank you very much for pointing out the additional references.
>
> `“The motivation behind Hyper-UT is not clear to me....It seems that it is simply yet another hypernetwork.”`
> As mentioned in the shared response, having an adaptive depth mechanism is a crucial part of this architecture. To reiterate the main point we are trying to make in this paper, we show how mechanism for adaptive depth and modularity can help transformers (1) generalize faster and better to examples of higher complexity (2) allocate compute more fairly.
> The particular design choice for the hyper module worked well empirically, and the precise overall scheme was intended to combine adaptive depth with modular reuse. The general intuition that leads to such architecture is that, we want a model that can allocate both type and amount of compute conditioned on the given example. From another point of view, we want to benefit from inductive biases of parameter sharing and recurrence in depth without sacrificing capacity.
>
> `“The experiment on C-PVR is not particularly convincing, as the task is a synethic one rather than a real problem. Could the authors find some other tasks that are more representative and yet realistic?”`
> That is a very good point. We think the experiments on the C-PVR task are still interesting, as it allows us to study the phenomena of interest more rigorously through controlled experiments.  The phenomena being how well the model can generalize to deal with examples that are more complex (requiring more sequential steps) than ones seen during training abstracting away from other factors that could influence the model. We also find it important to show this on more realistic use-cases. The main issue is that for real benchmarks it is really hard to come up with an automated way of annotating the complexity of the examples. Also, the space of potential solutions that the model can converge to that is not necessarily aligned with a defined complexity measure is much wider, making it very challenging to measure phenomena like generalization over complexity in these benchmarks. Nevertheless, we agree it is very important to build such benchmarks or find ways to show generalization over complexity on more realistic datasets.  We will discuss this point in the updated version of the paper. It is worth noting that we do have some experiment on ImageNet to show the generality of the suggested architecture.

---

### Official Review · Reviewer_tf4F · 2023-11-09

**Soundness:** 4 excellent
**Presentation:** 3 good
**Contribution:** 4 excellent
**Rating:** 5
**Confidence:** 4

**Summary:**

This paper focuses on the generalization performance of Transformers across various levels of task complexity. The paper introduces a new synthetic task, C-PVR (conditional point-value retrieval) to evaluate this generalization capability. In C-PVR, the model is asked to find a value at a specific position indicated by a pointer. In contrast to the PVR task (Zhang et al., 2021), C-PVR requires the model to navigate through multiple pointers until it reaches the desired position. The authors define the task's complexity in C-PVR by quantifying the number of hops to find the target value. Based on this task, the authors observe that modularity and adaptivity across tasks of varying complexity are significant to achieve better generalization. To address this, they propose Hyper-UT, a model equipped with Hyper-Modules that contains both modularity and adaptivity.
In the experiments, the authors demonstrate that Hyper-UT exhibits better generalization performance and efficiency when compared to conventional Transformers. Additionally, it performs competitively on ImageNet-1k, a well-known System-1 task. Furthermore, they analyze the generalization performance of pre-trained language models like T5 on C-PVR and observe that the scratch-pad can enhance the chain-of-thought ability of these models.

**Strengths:**

* They introduced a novel task that can evaluate the generalization capacity of Transformers across different complexity.
* They demonstrated that the generalization capacity of Transformers across task complexity can be improved through modularity and adaptivity.
* They compared and analyzed various models to confirm the importance of modularity and adaptivity in improving generalization capacity.
* The paper is well-written and easy to understand.

**Weaknesses:**

* In Section 4.2, a comparison with the T5 model trained from scratch is absent. To explore the influence of pre-training with language modeling on the generalization capacity, it is essential to include a comparison between the pre-trained T5 model and a T5 model trained from scratch specifically for the C-PVR task. This comparison will provide valuable insights into the impact of pre-training on the generalization.
* Comparing models with and without the scratch-pad in Figure 5 is challenging. It would be more helpful to combine Figure 5 (a) with Figure 5 (b), and Figure 5 (c) with Figure 5 (d).
* Ablation studies on the Hyper-Module are missing. To gain a better understanding of the effects of modularity and adaptivity, it would be beneficial to include an ablation study on the size of the weight embedding pool.

**Questions:**

* What do the notations "32x3" and "128x2" mean in Table 2 and Table 3 in the appendix?
* Is Hyper-UT also parameter-efficient compared to other methods?
* Is it possible to apply multi-head attention to the attention-based router? How it will affect the generalization capacity compared to the single-head one?

---

> ### Author Response · Authors · 2023-11-20
>
> Thank you for the thorough review and all the great questions and comments.
>
> `“A comparison with the T5 model trained from scratch is absent.”`
> We agree that the results with the T5 model are not directly comparable to transformers from scratch that we have in Figure 7. However, the main goal in these experiments was to show that the task remains challenging even for transformers  pre-trained on a language modeling objective. We will include experiments with the T5 architecture trained from scratch in the next version of the paper (the camera  ready version if the paper gets accepted). We speculate the results not to be very different than transformer encoders trained from scratch when there is no scratch-pad (Figure 7), since the decoding part would only have one decoding step.
> For the setting with scratch-pad we would expect better results (compared to the vanilla version) since the scratch-pad is basically augmenting the model with a mechanism for adaptive compute. While this effect has already been studied in other works, it would be interesting to see how scratch-pad or CoT helps in the context of the C-PVR task specifically (when there is no pretraining).
>
> `Comparing models in Figure 5.`
> We intentionally separated the plots to make them easier to read, but we agree that this makes the comparison a bit more difficult. We will update the plots and will incorporate this suggestion.
>
> `“What do the notations "32x3" and "128x2" mean in Table 2 and Table 3 in the appendix?"`
>  Apologies for the typo. It is meant to be 32x128 and 128x256. It is <number of model embeddings> \times <size of the embeddings>.
>
> `“Is Hyper-UT also parameter-efficient compared to other methods?”`
> Intuitively we think, at larger scales, Hyper-UT can also be more parameter-efficient but in this paper our main focus has been on the flop efficiency and we have not investigated this aspect. In the current experimental results in this paper, Hyper-UT has actually more parameters compared to the vanilla transformer / vision transformer and certainly universal transformer with full parameter sharing but it does converge to use fewer flops per example (more specifically less flops for simpler example and more flops for more complex examples).
>
> `“Is it possible to apply multi-head attention to the attention-based router? How it will affect the generalization capacity compared to the single-head one?”`
> Intuitively, this might help for more complicated tasks, but the effect might be less significant for a task like C-PVR. There might also be some interaction between the number of heads and the size of the embeddings. We will include an ablation experiment for the number of heads in the next version of the paper (So far we have not seen an improvement in efficiency or accuracy when increasing number of heads on the C-PVR task). Nevertheless, we think it is important to continue research in this direction to find the optimal and most efficient way of incorporating modularity into adaptive models.

---

> > ### Comment · Reviewer_tf4F · 2023-12-05
> > **Official Comment by Reviewer tf4F**
> >
> > I am grateful to the authors for addressing my comments. After going through the feedback from other reviewers and the discussions, I agree that the novelty of combining ACT with Hyper-Module seems a bit limited. However, I've realized that this paper introduces the importance of the multi-hop problem, which hasn't been explored much in other papers, and it's still important. Additionally, the paper suggests a new synthetic task to understand the multi-hop problem and what properties Transformers should have to solve it.
> > In that sense, I think the paper is valuable for revealing an important problem and its characteristics. Hence, I maintain my evaluation score for the paper.

---

### Author Response · Authors · 2023-11-20
**Shared Response (Thank you for all the comments and feedback)**

We would like to thank all the reviewers for reviewing our paper and for their comments and feedback.

Here we respond to the shared concerns and questions by the reviewers.  We will separately respond to the reviewers about the remaining points.

## Concerns about novelty (Reviewers PAjo, D45, Qbwm ):
We would like to re-iteraterate that the main goal in this paper is to emphasize the challenge of generalizing to examples that require more compute (in terms of number of sequential steps) than examples seen during training. We show that this is indeed not a trivial problem. Then, we show that augmenting models with adaptive time compute (such as universal transformer) with a mechanism for modularity leads to better generalization and efficiency. Both adaptive compute and modular compute are critical components of this idea and have been studied independently. We do not claim novelty in the specific means of realizing/implementing either adaptivity or modularity. We believe the missing piece of puzzle in this context was to apply these mechanism simultaneously to boost their effect. While we do get the best results when we use this specific parameterization of hyper-networks for example compared to using FiLM layers, in principle the concept applies to any strategy to modularize the model. Indeed, we think there is still a lot to do here for future research figuring out the most efficient way to do so.

## Concerns about ablation experiments (Reviewers tf4F, D459):
We agree with the reviewers that proposal of the hyper-ut architecture would be more convincing with some ablation experiments on the architecture. The primary objective of our paper was to emphasize assessing generalization with respect to the complexity of examples and the efficiency/fairness of allocation of compute. We aimed to show how augmenting adaptive models with modularity could make them more effective in these scenarios and also in general. While we chose not to overly concentrate on the implementation details of the hyper-ut, we acknowledge that the paper could be enhanced by including ablation experiments, such as those examining the size and number of module embeddings and we plan to include these in the next version of the paper (the camera ready version if the paper is accepted).
(So far, our result suggest increasing the number and size of experts impact the convergence speed and the efficiency in terms of maximum number of steps (layers) per example.)

---

### Meta-Review · Area_Chair_kxG1 · 2023-12-07

**Metareview:**

This paper investigates the impact of adaptive computation and modality in Transformers for out-of-length generalization. To this end, the authors employed simple modifications to the Transformers based on prior works to adaptively increase the network depth per data and employed shared parameters across layers through the hyper-network. The proposed method was evaluated on a synthetic dataset that requires multi-hop reasoning on simple sequences, and demonstrated improved performance over plain Transformers.

The paper received three borderline rejects and one accept. The major concerns raised by the reviewers were about (1) limited technical contribution, since the components to equip adaptive computation and modularity are largely from the prior works, (2) lacking results on realistic setup, since all experiments on multi-hop reasoning are conducted in a rather simple synthetic dataset, and (3) missing ablation study.

After reading the paper, reviews, and rebuttal, AC concluded that the results presented in the paper provide valuable insights, yet are insufficient to generalize the authors’ claim. Although AC understands that the synthetic task is useful to directly evaluate the performance vs. task complexity, it is an overly simplified environment. Perhaps it is not surprising that Transformers with appropriate bottleneck structures over depth and parameter structure are favored over naive Transformers i.e., it is not easy to specify whether the source of improvement is from the right choice of expressiveness bottleneck or from adaptive reusability structure of parameters. Also, it is not evident from the results that whether we can generalize the insights into realistic settings (e.g., large-scale models). Typically, the foundation models are trained on massive web-scale data, and internally learn reusable structures i.e., some degree of modularity emerges implicitly. It is then not clear that employing the explicit reusable structure will be still useful in such cases. To understand this, it would be valuable to have additional evaluation on more realistic datasets, such as multi-hop question answering.

**Justification For Why Not Higher Score:**

Experiment results are insufficient to generalize the authors' claims to a broad class of realistic tasks.

**Justification For Why Not Lower Score:**

N/A

---

### Decision · Program_Chairs · 2024-01-16

Reject